# Cardiotoxicity Associated with Gemcitabine: Literature Review and a Pharmacovigilance Study

**DOI:** 10.3390/ph13100325

**Published:** 2020-10-21

**Authors:** Marc Hilmi, Stéphane Ederhy, Xavier Waintraub, Christian Funck-Brentano, Ariel Cohen, Aurore Vozy, Bénédicte Lebrun-Vignes, Javid Moslehi, Lee S. Nguyen, Joe-Elie Salem

**Affiliations:** 1Department of Pharmacology, Regional Pharmacovigilance Centre, Pitié-Salpêtrière Hospital, INSERM CIC-1901, Sorbonne Université, AP-HP, 75006 Paris, France; hilmi.marc@gmail.com (M.H.); christian.funck-brentano@aphp.fr (C.F.-B.); aurore.vozy@aphp.fr (A.V.); benedicte.lebrun-vignes@aphp.fr (B.L.-V.); nguyen.lee@icloud.com (L.S.N.); 2Department of Cardiology, Saint-Antoine Hospital, Sorbonne Université, AP-HP, 75006 Paris, France; stephane.ederhy@aphp.fr (S.E.); ariel.cohen@aphp.fr (A.C.); 3UNICO-GRECO APHP.Sorbonne Cardio-Oncology Program, Sorbonne Université, 75006 Paris, France; 4Department of Cardiology, Pitié-Salpêtrière Hospital, Sorbonne Université, AP-HP, 75006 Paris, France; xavier.waintraub@aphp.fr; 5EA Epiderme-Epidemiology in Dermatology and Evaluation of Therapeutics, Université Paris-Est Créteil, 94000 Créteil, France; 6Division of Cardiovascular Medicine, Cardio-Oncology Program, Vanderbilt University Medical Center, Nashville, TN 37232, USA; javid.moslehi@vumc.org; 7Research and Innovation of CMC Ambroise Paré, 92200 Neuilly-sur-Seine, France

**Keywords:** gemcitabine, pericarditis, myocardial ischemia, heart failure, arrhythmias, cardio-oncology

## Abstract

Background: Gemcitabine is a nucleoside analog, widely used either alone or in combination, for the treatment of multiple cancers. However, gemcitabine may also be associated with cardiovascular adverse-drug-reactions (CV-ADR). Methods: First, we searched for all cases of cardiotoxicity associated with gemcitabine, published in MEDLINE on 30 May 2019. Then, we used VigiBase, the World Health Organization’s global database of individual case safety reports, to compare CV-ADR reporting associated with gemcitabine against the full database between inception and 1 April 2019. We used the information component (IC), an indicator value for disproportionate Bayesian reporting. A positive lower end of the 95% credibility interval for the IC (IC_025_) ≥ 0, is deemed significant. Results: In VigiBase, 46,898 reports were associated with gemcitabine on a total of 18,908,940 in the full database. Gemcitabine was associated with higher reporting for myocardial ischemia (MI, n: 119), pericardial diseases (n: 164), supraventricular arrhythmias (SVA, n: 308) and heart failure (HF, n: 484) versus full database with IC_025_ ranging between 0.40 and 2.81. CV-ADR were associated with cardiovascular death in up to 17% of cases. Conclusion: Treatment with gemcitabine is associated with potentially lethal CV-ADRs, including MI, pericardial diseases, SVA and HF. These events should be considered in patient care and clinical trial design.

## 1. Introduction

Gemcitabine is a cytidine analog that incorporates into DNA and terminates chain elongation by inhibiting ribonucleotide reductase and DNA repair [1]. It is still widely used, either alone or in combination to treat various cancers including lung, pancreas, bladder, breast, ovary and bile duct carcinomas, lymphomas and uterine sarcomas [2]. Gemcitabine is generally preferred in elderly or fragile patients due to lower toxicity profile compared to other anticancer drugs. While myelosuppression is the most commonly observed adverse drug reaction (ADR) associated with this molecule, several other ADRs have emerged since gemcitabine was approved by the Food and Drug Administration (FDA) [2], including thrombotic microangiopathy [3], interstitial pneumonitis [4] and capillary leak syndrome (CLS) [5].

According to the FDA label, incidence of cardiovascular ADR (CV-ADR) associated with gemcitabine is low and rarely leads to drug discontinuation [2]. The European Medicines Agency also notifies the particular caution with patients presenting a history of cardiovascular events due to the risk of CV-ADR with gemcitabine [6].

We identified three patients in Saint-Antoine and Pitié-Salpêtrière hospitals (Paris, France) who developed a CV-ADR suspected to be related to gemcitabine. They presented with pancreatic cancer and CV-ADR were pericardial effusion associated with heart failure in two of them, which occurred 6 to 8 months after gemcitabine initiation. Partial or complete recovery was observed after instauration of heart failure therapeutics (see details of in Table 1). One patient presented its pericardial effusion as part of a CLS recovered after glucocorticosteroids administration (Appendix A present dynamics of his recovery).

Herein, we aim to further delineate the overall spectrum of CV-ADR associated with gemcitabine. First, we present three news cases of cardiotoxicity associated with gemcitabine (Table 1) and their management (i), we perform a literature review focusing on the description of CV-ADR reported on gemcitabine in MEDLINE (ii) and (iii) we use VigiBase, the WHO’s international pharmacovigilance database of individual case safety reports (reports, thereafter) to describe the reported CV-ADR cases associated with gemcitabine.

## 2. Results

### 2.1. Literature Review

Twenty-three cases reporting gemcitabine-associated CV-ADR were retrieved from literature review including myocardial ischaemia (MI, n: 4), heart failure (HF, n: 10), supraventricular arrhythmias (SVA, n: 6), and pericardial disorders (n: 8), 2 of which were in a context of CLS. Table 1 describes the main characteristics and outcome of these patients. Patients were treated with gemcitabine for pancreatic or lung cancer and lymphoma in respectively, n: 12/23, 52%, n: 5/23, 22%, and n: 5/23, 22%. All patients needed in-hospital management for these CV-ADR and 10/23 (44%) patients did not fully recovered. HF patients presented with altered left ventricular ejection fraction in n: 7/10, 70%; and recovered partially from HF symptoms in n: 6/10, 60%, and completely in n: 4/10, 40% after adequate treatment (i.e., diuretics, betablockers and angiotensin-converting enzyme inhibitors). Three out of 8 patients (38%) with pericardial disorders evolved into constrictive pericarditis. The two patients with pericardial effusion in a context of CLS completely recovered after pericardiocentesis or glucocorticosteroids. Gemcitabine was definitely stopped in 16/23 cases (70%) and 7 patients were rechallenged. Recurrence of CV-ADR after rechallenge occurred in 4/7 cases (57%). Of note, SVA and MI appeared early (i.e., hours/days) after gemcitabine infusion whereas HF occurred later (i.e., weeks/months).

Moreover, among the 106 randomized clinical trials evaluating gemcitabine as monotherapy (overall number of patients = 14015), 17 trials (*n* = 2386) had reports for at least one CV-ADR in the published work (Appendix A for the flow chart of selected trials). In these latter 17 trials, 33 CV-ADR were reported, leading to an estimate of CV-ADR incidence rate on gemcitabine monotherapy ranging from 0.24% (33/14015) to 1.38% (33/2386), varying with the denominator considered. Quality of CV-ADR reporting in these trials, mostly from decades ago was too low to make sure that all CV-ADR events were effectively captured and reported in the publication. These events were severe in n: 27/33 (82%) (n: 14/33 grade 3/4 and n: 13/33 death), and included 8 MI, 2 pericardial effusions, 7 HF and 1 arrhythmia (Table 2).

### 2.2. Pharmacovigilance Study

The total number of ADR reported with gemcitabine was 46,898 and the total number of ADR in VigiBase was 18,908,940 on 1 April 2019. We identified four broad cardiovascular entities totalizing 973 reports for which reporting was significantly increased with gemcitabine (IC_025_ > 0) versus full database (Table 3): pericardial diseases (n: 164); MI (n: 119); SVA (n: 308) and HF (n: 484). Sub-classifications and intersections between these CV-ADRs is shown in Appendix A. Other CV-ADR including myocarditis, conduction disorders, ventricular arrythmias, long QT, cardiac arrest and valve disorders were not over-reported with gemcitabine (Figure 1). The most overlapping CV-ADRs were HF within pericardial reports (33/164, 20%) vs. HF within SVA and MI reports (36/308, 12%; 5/119, 4%; *p* < 0.001; respectively). Otherwise, these conditions were moderately overlapping (0–7%, Figure 1, Table 4).

We described the main characteristics of these 973 reports as a function of each type of CV-ADR and concurrent reported conditions in Table 4 and Appendix A. These cases were declared worldwide mainly from healthcare professionals (90–97%) in post-marketing setting (89–94%), affecting mainly adults (range: 16–89 yo) and increasingly over decades starting in 1997. Indications were mainly for pancreatic (25–39%), and lung (24–36%) cancer. Notably, the third most represented cancer was urothelial for MI (13/93, 14%), bile duct for SVA (21/228, 9%), lymphoma for pericardial diseases (19/97, 20%), and breast cancer for HF reports (36/293, 12%; Table 4, *p* < 0.0001).

Male were more affected by gemcitabine-associated MI (65/105, 62%), SVA (184/295, 62%), HF (235/451, 52%) while women were over-represented in pericardial reports (95/152, 62%; *p* < 0.0001). CV-ADR were severe in the majority of cases (76–94%), with cardiovascular related death particularly predominant in MI (20/115, 17%) and HF (83/477, 17%) vs. SVA (17/289, 6%) and pericardial reports (5/144, 3%, *p* < 0.0001; Appendix A). Accordingly, when outcome was known, complete recovery was more prevalent in pericardial (48/55, 87%) and SVA (107/132, 81%) vs. MI (35/51, 69%) and HF (115/163, 70%) reports (p:0.02). Median time to onset for HF (75 days, IQR [22, 166]) was longer vs. other CV-ADRs (MI: 29 days, IQR [6, 80]; SVA: 14 days, IQR [4, 52]; pericardial: 55 days, IQR [13, 144], *p* < 0.0001, Figure 2). Gemcitabine was the only suspected liable drug in patients with pericardial diseases (124/164, 76%) more often than in other CV-ADRs (503/809, 62%; *p* = 0.001, Table 4). The analysis of co-reported drugs showed that platins were overrepresented in MI reports (55/119, 46%) vs. within other CV-ADRs (209/854, 25%, *p* = 0.04). Prevalence of other concomitant anticancer drugs intake as a function of type of gemcitabine associated CV-ADRs are presented in Table 4. Combination therapy (gemcitabine and at least one other anticancer drug suspected) was more incriminated than gemcitabine alone in myocardial infarction reports (86/119, 72% vs. 33/119, 28%). A significant association was detected for HER-2 blockers and HF (*p* = 0.02), taxanes and SVA (*p* < 0.0001), immune-checkpoint-inhibitors and pericardial disorders (*p* = 0.001), epidermal growth factor receptor blockers and/or platins and MI (*p* = 0.001)—see Table 4 for more details.

Patients with pericardial diseases were younger than those with other CV-ADRs (55.33 ± 15.6 vs. 65.00 ± 10.7 years, *p* < 0.001). As compared to other CV-ADRs, pericardial diseases were associated with more concurrent CLS (4/164, 2% vs. 5/809, 0.01%; *p* = 0.03), pleural effusion (56/164, 34% vs. 14/809, 2%; *p* < 0.0001), ascites (8/164, 5% vs. 7/809, 0.01%; *p* = 0.001), edema (27/164, 17% vs. 74/809, 9%; *p* = 0.005) and radiation recall reaction (17/164, 10% vs. 0/809, 0%, *p* < 0.0001). Lastly, MI patients were associated with more concurrent stroke versus those with other CV-ADRs (10/119, 8% vs. 16/854, 2%; *p* < 0.0001).

## 3. Discussion

We report the first large-scale analysis associating specific CV-ADR with gemcitabine. This study of individualized reportable events from the WHO pharmacovigilance database combined with the literature review allowed us to better characterize the CV-ADR associated with gemcitabine, notably the clinical characteristics including time to onset and severity of approximatively 1000 reports with gemcitabine-associated cardiotoxicity, versus few isolated case reports published previously (Table 1).

Gemcitabine was associated with MI, SVA, HF and pericardial diseases. The same cardiotoxicity signals from gemcitabine have also been reported in clinical trials. Aapro et al. pooled 979 patients treated by gemcitabine in 22 phase-2 trials and showed that incidence of MI, HF, arrhythmias and pericarditis were 0.5%, 0.4%, 0.2% and 0.1%, respectively [38]. Our meta-proportion analysis of randomized clinical trials evaluating gemcitabine as monotherapy showed similar incidence estimates of ~1% for overall CV-ADR including MI, SVA, HF and pericardial diseases. We showed that mortality associated with these CV-ADRs ranged from 3% for after pericardial diseases versus 17% for after MI and HF. Severity including death and grade 3/4 events reported in randomized clinical studies (82%) was similar to that of the pharmacovigilance reports (87%). Though, it has to be noted that the quality of CV-ADR adjudication in the trials considered in our analysis was low precluding the possibility of precisely characterizing these CV-ADR. Pericardial diseases had the strongest association with gemcitabine administration (highest IC_025_, Table 3), and were presented as cardiac tamponade (28/164, 17%) or constrictive pericarditis (6/164, 4%). Noteworthily, pericardial disorders could be part of CLS, a systemic disease determined by vascular protein leakage and diffuse serosa effusions requiring specific management, including glucocorticoids [39,40,41]. Association between gemcitabine and CLS using VigiBase have been described previously [5].

Although CV-ADRs were severe, required hospitalization and often were not fully reversible, cardiac toxicity of gemcitabine is not well known to clinicians. Even in the absence of cardiovascular risk factors, cardiotoxicity related to gemcitabine should be immediately suspected in the case of breathlessness, palpitations, or chest discomfort. Moreover, cardiovascular screening (e.g., echocardiography assessing pericardium and left ventricular ejection fraction, past cardiac history and prior radiation therapy) may identify patients at higher risks of gemcitabine-associated cardiotoxicity but this strategy needs to be evaluated in a dedicated study. Furthermore, in some case-reports (Table 1), gemcitabine re-challenge led to recurrence of the CV-ADR suggesting that decision to restart gemcitabine needs to be weighted between cardiotoxicity and anti-tumor efficacy [42].

Several hypotheses are suggested to explain association of gemcitabine with the various CV-ADRs, but no thorough preclinical mechanistic study is available. Similar to fluoropyrimidines (other antimetabolites), vasospasm has been proposed to be responsible for gemcitabine-associated MI [7,8,9]. Interestingly, MI associated with gemcitabine were often co-reported with strokes in our study, suggesting that shared cardiovascular risk factors and/or a shared pathophysiological mechanisms (i.e., arterial vasospasm) may play a role. For pericarditis, Vogl et al. highlighted for the first time an association between gemcitabine and pericarditis via a radiation recall reaction (acute inflammatory reaction confined to previously irradiated areas triggered when chemotherapy agents are administered) [19]. The recalled inflammation induced by gemcitabine may lead to fluid accumulation in the incompliant pericardial space and ultimately tamponade [43]. Consistently with this hypothesis, we found an association between radiation recall reaction and pericardial diseases. However, prior irradiation was not mandatory for gemcitabine induced pericardial effusion (3 with no prior irradiation among 8 cases with gemcitabine pericarditis, Table 1). For HF, a retrospective study suggested diabetes, coronary artery disease and a total gemcitabine dose >17.000 mg/m^2^ as risk factors for developing gemcitabine-induced HF but validity of results are limited by a small sample size (7 HF on a total of 156 gemcitabine treated patients) [44]. In VigiBase, we were unable to obtain cumulative doses of gemcitabine but our data showed increased time to onset and increased duration of gemcitabine treatment for HF vs. other CV-ADRs patients, further supporting this cumulative dose effect. Notably, we found that specific concomitant intake of anticancer drugs on top of gemcitabine were more likely to be associated with different type of cardiotoxicities such as platins with MI, HER-2 blockers with HF, and immune checkpoint inhibitors with pericardial disorders. Indeed, these drugs are known to induce these conditions and the hypothesis of toxic synergy or multiple hit mechanisms with these combinations is likely [18,45,46]. Overall, the co-administration of drugs that can induce CV-ADR should strengthen the surveillance of high-risk cardiovascular patients receiving combination of such agents.

We acknowledge several limitations of VigiBase pharmacovigilance analysis, the first of which is under-reporting of suspected CV-ADRs. While the accurate magnitude of underreporting cannot be computed, estimates vary up to 90% of the actual adverse events not being reported [47,48]. However, VigiBase is a worldwide database, gathering data from over 130 countries and with almost 1000 CV-ADR reports related to gemcitabine vs. few case-reports previously described. Sources of reports are non-homogeneous with limited possibility for verification of the clinical, laboratory tests, or radiological findings and re-assessment of the causality of the drug-ADR combination. Thus, in approximately two-third of reports, gemcitabine was the only suspected drug for the reported cardiotoxicity. The exact denominator of patients exposed to gemcitabine cannot be evaluated precluding estimation of incidence of these CV-ADRs using VigiBase. Instead, total number of reports for the studied drug is used as denominator for this kind of analysis [45]. The value of disproportionality analysis for CV-ADR associated with anticancer drugs has already been shown in various settings with confirmation of signals by prospective trials or basic mechanistic studies, such as with myocarditis induced by immune-checkpoint inhibitors, or QT prolongation induced by anti-hormones but nevertheless, there is still a risk that results from pharmacovigilance databases might be misleading [45,49,50]. Yet, these above-described CV-ADRs associated with gemcitabine must warrant caution.

## 4. Materials and Methods

### 4.1. Literature Review

We performed a systematic search on PubMed (MEDLINE) using the terms “Cardiotoxicity AND Gemcitabine” with no filter, at the date of 30 May 2019.Then, we screened the bibliography of selected publications. We identified a total of 18 publications including case-reports between 1999 and 2018 focusing on gemcitabine cardiotoxicity. We also performed a systematic search on Pubmed (MEDLINE) for randomized clinical trials using the term “gemcitabine” up to 30 September 2020 and found 958 manuscripts. We then selected articles evaluating gemcitabine monotherapy in at least one of the treatment arms and found 106 studies (Appendix A).

### 4.2. Pharmacovigilance Study

#### 4.2.1. Study Design and Data Sources

This observational, retrospective, pharmacovigilance study is based on ADR reported in deduplicated VigiBase, the WHO’s international database of individual case safety reports (ICSRs, or reports in the text) which includes reports from over 130 countries [51]. These reports originate from different sources, such as healthcare professionals, patients, and pharmaceutical companies, and are generally notified post-marketing. The use of confidential, electronically processed patient data was approved by the French National Commission for Data Protection and Liberties (reference number #1922081). It was not appropriate or possible to involve patients or the public in the design, or conduct, or reporting, or dissemination plans of our research. ClinicalTrials.gov Identifier: NCT03530215

#### 4.2.2. Procedures

This study included all possible CV-ADR according to Preferred Term (PT) levels of the Medical Dictionary for Regulatory Activities (MedDRA; version 21.1, the International Council for Harmonisation of Technical Requirements for Pharmaceuticals for Human Use (ICH), McLean, VA 22102, USA) within the group query of System Organ Class: cardiac disorders [52] between inception (14 November 1967) through 1 April 2019. Cardiac disorders evaluated included cardiac arrhythmias, neoplasms, valve disorders, congenital disorders, heart failures, coronary artery, myocardial, pericardial and endocardial disorders. CV-ADR specifically assessed in the analysis were those notified as suspected to be induced by gemcitabine. Each report contains general administrative information (country, date, and reporter qualification), patient characteristics (sex and age), drugs (indication, start and end dates), and reactions (onset and end date, seriousness, outcome).

#### 4.2.3. Statistical Analysis

VigiBase allows for disproportionality analysis (also known as case–non-case analysis), which we used to assess whether suspected drug-induced CV-ADR were differentially reported with gemcitabine versus the full database. Disproportionality analysis compares the proportion of selected specific ADR (e.g., pericarditis) reported for a single drug (e.g., gemcitabine) with the proportion of the same ADR for a control group of drugs (e.g., full database). The denominator in these analyses is the total number of ADR reported for each group of drugs. If the proportion of ADR is greater in patients exposed to a specific drug (cases) than in patients not exposed to this drug (non-cases), then an association can be made between the specific drug and the reaction leading to a potential safety concern. Disproportionality in VigiBase is generally calculated using the information component (IC), an indicator value for disproportionate Bayesian reporting when using the full database as comparator [45]. Disproportionality can also be calculated by using a frequentist disproportionality estimate, i.e., the reporting odds ratio (ROR) [49,53].

Calculation of the IC using a Bayesian confidence propagation neural network was developed and validated by the Uppsala Monitoring Centre (organism managing VigiBase) as a flexible, automated indicator value for disproportionate reporting that compares observed and expected drug–ADR associations to find new drug–ADR signals with identification of probability difference from the background data (full database) [54]. Probabilistic reasoning in intelligent systems (information theory) has proved to be effective for the management of large datasets, is robust in handling incomplete data, and can be used with complex variables [54]. Several examples of validation with the IC exist, showing the power of the technique to find signals soon after drug approval by a regulatory agency (e.g., an association between captopril and coughing), and to avoid false positives, whereby an association between a common drug and a common ADR occurs in the database only because the drug is widely used and the ADR is frequently reported (e.g., between digoxin and rash) [54,55].

The statistical formula is as follows:$$\mathrm{IC}=\log2\left[\frac{\left(Nobserved+0.5\right)}{\left(Nexpected+0.5\right)}\right]$$
where
$$Nexpected=\left[\frac{\left(Ndrug\;\times \;Neffect\right)}{Ntotal}\right]$$

*Nexpected* is the number of ICSRs expected for the drug–ADR combination. *Nobserved* is the actual number of ICSRs for the drug–ADR combination. *Ndrug* is the number of ICSRs for the drug, regardless of ADR. *Neffect* is the number of ICSRs for the ADR, regardless of drug. *Ntotal* is the total number of case reports in the database. IC_025_ is the lower end of a 95% credibility interval for the IC. A positive IC_025_ value (>0) is the traditional threshold deemed significant [45,54,55]. Characteristics of cases were described in terms of means (standard deviation, SD) or medians (interquartile range, IQR) for quantitative variables, and in terms of numbers and proportion for qualitative ones. Unpaired Student *t*-tests were performed to compare means of two groups and analysis of variance (ANOVA) or Kruskal–Wallis test was used to compare means or medians of more than two groups, respectively. χ^²^-tests were performed for comparisons of qualitative variables. *p*-values < 0.05 were deemed significant.

## 5. Conclusions

In conclusion, this international global pharmacovigilance study showed that gemcitabine is significantly associated with an over-reporting versus full database of drug-induced myocardial ischemia, pericardial diseases, supraventricular arrhythmias and heart failure with almost one thousand cases and moderate overlap between these conditions (0–20%). These conditions were described rarely in the past literature, with only 20 case-reports. Pericardial effusion associated with gemcitabine were eventually part of a capillary leak syndrome, a systemic condition responsive to glucocorticoids. These cardiovascular adverse events remained rare (incidence ~≤ 1%) and generally occurred within 1–2 months of gemcitabine start, except heart failure adverse events which occurred later and showed less reversibility (30% sequalae or no recovery). Subsequent cardiovascular death occurring in 3–17% of case, emphasizing their seriousness. Hence, cardiotoxicities related to gemcitabine are potentially life-threatening should be investigated in patients care and clinical trials design, particularly in combination with other cardiotoxic drugs.

## Figures and Tables

**Figure 1 pharmaceuticals-13-00325-f001:**
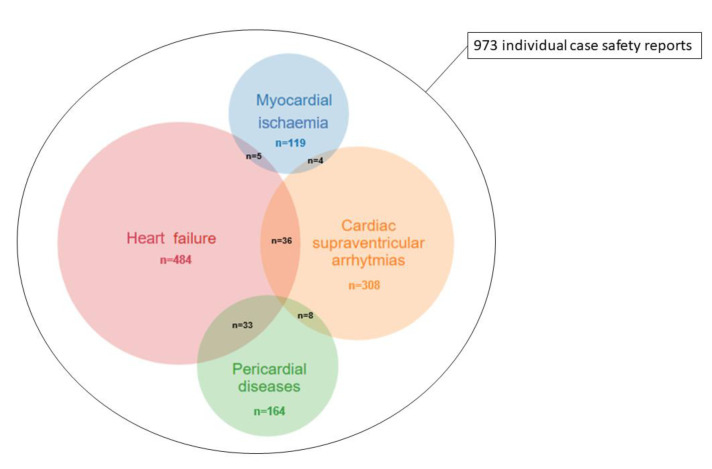
Individual case safety reports and overlap for gemcitabine-associated cardiovascular adverse drug reactions (CV-ADRs) in VigiBase (accessed on 1 April 2019).

**Figure 2 pharmaceuticals-13-00325-f002:**
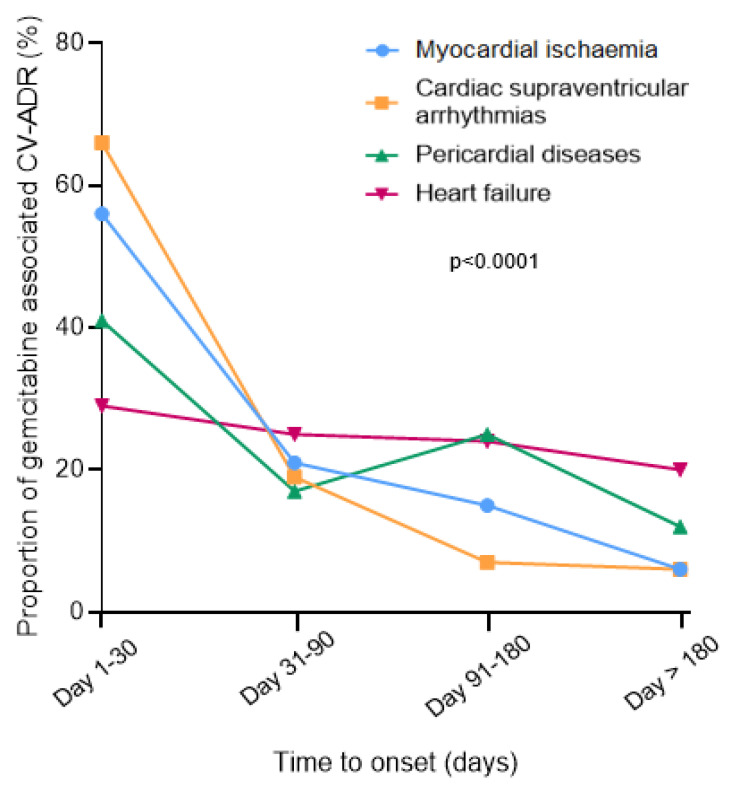
Time to event onset for gemcitabine-associated cardiovascular adverse drug reactions (CV-ADRs) in VigiBase (accessed on 1 April 2019). The Chi^2^ test was used to generate the *p*-value.

**Table 1 pharmaceuticals-13-00325-t001:** Case reports of cardiotoxicity associated with gemcitabine in MEDLINE (thru 30 May 2019).

Author, Year, Reference	Cardiovascular Adverse events	Age, Gender Type of Cancer	Cardiovascular Risk Factors and Medical History	Gemcitabine Dosing/Cycle Cumulative Doses	Time to Onset after 1st Intake; after Infusion	Concurrent Suspected Drugs	Management	Outcome	Rechallenge
Ozturk et al., 2009, [7]	Acute myocardial infarction	59, Female Leiomyosarcoma	HTN, Dyslip, diabetes, CAD	900 mg/m^2^ D1-8-21 (D1 = D21); 1800 mg/m^2^	8 days; 30 min	Docetaxel	Aspirin, clopidogrel, BB-, heparin, nitrate revascularization	Discharged D2, Complete recovery	No
Bdair et al., 2006, [8]	Acute myocardial infarction; cardiac arrest (ventricular tachycardia)	43, Female Lung cancer	HTN, smoker Postpartum cardiomyopathy, stroke, CAD	1000 mg/m^2^ D1-8-21 (D1 = D21); 4000 mg/m^2^	42 days; 72 h	No	Aspirin, BB-heparin, nitrate, glycoprotein IIb/IIIa inhibitors	Discharged D4, Complete recovery	No
Kalapura et al., 1999, [9]	Acute myocardial infarction; HF (LVEF:45%)	54, Male Pancreatic cancer	None	NA; 9500 mg	60 days; 6 h	No	Aspirin, heparin, BB-	Discharged D7, Partial recovery	Yes, recurred on nitrate and BB-
Santini et al., 2000, [10]	AF	78, Male Pancreatic cancer	Paroxysmal AF	NA D1-8-15-28 (D1 = D28); NA	18 h; 18 h	No	Propafenone	Discharged within D, Complete recovery	Yes, recurred on propafenone
Ferrari et al., 2006, [11]	AF	72, Female Lung cancer	None	1200 mg/m^2^ D1-8-21 (D1 = D21); 1200 mg/m^2^	18 h; 18 h	No	Amiodarone	Discharged D1, Complete recovery	No
Ferrari et al., 2006, [11]	AF	73, Female Lung cancer	None	1200 mg/m^2^ D1-8-21 (D1 = D21); 7200 mg/m^2^	42 days; 12 h	No	Digoxin	Discharged D5, Partial recovery (AF rate control)	No
Tavil et al., 2007, [12]	AF	65, Male Lung cancer	None	1200 mg/m^2^ D1-8-21 (D1 = D21); 2400 mg/m^2^	8 days; 7 h	Cisplatin	Propafenone, verapamil	Discharged D2, Complete recovery	No
Ciotti et al., 1999, [13]	AF	70, Male Pancreatic cancer	None	NA; NA	6 days; 6 days	No	Digoxin	Complete recovery after 12 days	Yes, recurred
Tayer-shifman et al., 2009, [14]	Junctional tachycardia (nodal reentrant)	67, Female Breast cancer	None	1000 mg/m^2^ D1-8-21 (D1 = D21); 3000 mg/m²	21 days; few hours	No	Adenosine, verapamil, BB-	Discharged D5, Complete recovery	No
Khan et al., 2014, [15]	HF (LVEF: 20%)	56, Male Pancreatic cancer	None	1000 mg/m^2^ D1-8-15-28 (D1 = D28); 6000 mg/m^2^	56 days; NA	No	Furosemide, BB-, ACE	Discharged D2, Partial recovery (LVEF 40% few months later)	Yes, recurred
Yajima et al., 2004, [16]	HF	82, Female Pancreatic cancer	NA	NA; 16,800 mg	2 years; NA	No	NA	Partial recovery	No
Alam et al., 2018, [17]	HF (LVEF: 40%) Myocardial ischaemia	62, MalePancreatic cancer	CAD, HTN	1000 mg/m^2^ D1-8-15-28 (D1 = D28); 13,000 mg/m^2^	112 days; NA	No	Diuretics	Discharged after weeks, Partial recovery (LVEF:40%)	No
Alam et al., 2018, [17]	HF (LVEF: 38%)	63, Male Pancreatic cancer	None	1000 mg/m^2^ D1-8-15-28 (D1 = D28); 7000 mg/m^2^	56 days; NA	No	Diuretics	Complete recovery (LVEF:67%)	No
Alam et al., 2018, [17]	HF (LVEF: 60%)	72, Female Pancreatic and lung cancer	HTN, diabetes, Dyslip, ex-smoker	1000 mg/m^2^, NA	28 days; NA	No	Diuretics	Complete recovery after few months	No
Mohebali et al. 2017, [18]	HF (LVEF: 20%)	67, Female Lymphoma	Dyslip	NA; NA	30 days; NA	Rituximab Oxaliplatin	Diuretics, ACE, BB-	Partial recovery at 6 months (LVEF:40%)	No
Hilmi et al., 2020, [NA]	HF (LVEF: 35%) Pericardial effusion	67, Female Carcinoma of Vater’s papilla	HTN, Dyslip	800-1000 mg/m^2^ D1-8-15-28 (D1 = D28); 14,800 mg/m^2^	170 days; 2 days	No	Furosemide, amlodipine, ACE, Pericardial tap	Discharged D9, Partial recovery at 1 year (LVEF:40%)	No
Hilmi et al., 2020, [NA]	HF (LVEF:20%) Pericardial effusion	47, Male Pancreatic cancer	Cardiac XR, previously treated with anthracyclines	1000 mg/m^2^ D1-8-15-28 (D1 = D28); 18,000 mg/m^2^	175 days; 7 days	No	Furosemide, BB-, ACE	Discharged D7, Complete recovery at 1 year	No
Hilmi et al., 2020, [NA]	Pericardial effusion CLS	71, Female Pancreatic cancer	None	1000 mg/m^2^ D1-8-15-28 (D1 = D28); 24,000 mg/m^2^	246 days; 6 days	No	Glucocorticoid, furosemide	Discharged D10, Complete recovery at 1 year	No
Vogl et al., 2005, [19]	Pericardial effusion	26, Female Lymphoma	Cardiac XR, and previously cisplatin/cytarabine	750 mg/m^2^; 750 mg/m^2^	1 day; 1 day	Rituximab Vincristine	Pericardial surgery (pericardial window)	Not recovered (constriction)	Yes
Vogl et al., 2005, [19]	Cardiac tamponade	36, Male Lymphoma	Cardiac XR, and previously AC	1000 mg/m^2^; 1000 mg/m^2^	3 days; 3 days	Rituximab Vincristine	Pericardial surgery (pericardial window)	Complete recovery 2 months later	No
Vogl et al., 2005, [19]	Pericardial effusion	53, Male Lymphoma	Cardiac XR, and previously AC	750 mg/m^2^ D1-14 (D1 = D14), 4500 mg/m^2^	70 days; NA	No	Glucocorticoid	Not recovered (constriction)	Yes
Vogl et al., 2005, [19]	Constrictive pericarditis	31, Female Lymphoma	Cardiac XR, and previously AC	1000 mg/m^2^ D1-14 (D1 = D14), 4000 mg/m^2^	30 days; NA	No	NA	Not recovered (constriction)	Yes
Kido et al., 2012, [20]	Pericardial effusion, CLS HF (LVEF:59%)	56, Female Pancreatic cancer	None	NA	120 days; NA	No	Furosemide, pericardial surgery (pericardiocentesis)	Discharged D20, Complete recovery (LVEF:76%)	No

Abbreviations: AC: anthracyclines/cyclophosphamide; ACE, angiotensin-converting enzyme inhibitor; AF, atrial fibrillation; BB-, betablockers; CAD, coronary artery disease; CLS, capillary leak syndrome; D, day; DM, diabetes; Dyslip, dyslipidemia; HF, heart failure; HTN, hypertension; LVEF, left ventricular ejection fraction; NA, not available; XR: radiotherapy.

**Table 2 pharmaceuticals-13-00325-t002:** Cardiovascular adverse drug reactions in randomized clinical trials evaluating gemcitabine in monotherapy with at least one report for cardiovascular adverse-drug-reactions (CV-ADR).

Study	Type of Study	Number of Patients in the Gemcitabine Monotherapy Arm	Type of Cancer	Median Follow-up	Number of Previous Chemotherapies	Cardiovascular Adverse Drug Reaction (CV-ADR)
Konstantinopoulos et al. [21]	II	36	Ovary	13.3 months	>1	Myocardial infarction: 1 grade 3
Conroy et al. [22]	III	169	Pancreas	26.6 months	0	Heart failure: 1 death
Melisi et al. [23]	II	52	Pancreas	Not available	0	Pericardial effusion: 1 death
Middleton et al. [24]	II	70	Pancreas	24.9 months	0	Myocardial infarction: 1 death
Neoptolemos et al. [25]	III	366	Pancreas	43.2 months	0	Cardiac disorders: 1 death, 1 grade 3, 4 grade 2
Evans et al. [26]	II	102	Pancreas	Not available	>1	Cardiac arrest: 3 deaths
Rougier et al. [27]	III	275	Pancreas	7.9 months	0	Heart failure: 1 grade 3
Gonçalves et al. [28]	III	52	Pancreas	27.7 months	0 or more	Cardiac disorders: 6 grade 3/4
Loehrer et al. [29]	II	35	Pancreas	Not available	0	Myocardial infarction: 1 death
Colucci et al. [30]	III	199	Pancreas	38.2 months	0	Arrhythmia: 1 grade 3
Richards et al. [31]	II	39	Pancreas	Not available	0	Myocardial infarction: 1 death
Spano et al. [32]	II	31	Pancreas	Not available	0	Myocardial infarction: 1 grade 3
Herrmann et al. [33]	III	156	Pancreas	Not available	0 or 1	Myocardial infarction: 1 death
Van Cutsem et al. [34]	III	347	Pancreas	Not available	0 or 1	Heart failure: 2 grade 2 Pericardial effusion: 1 death Myocardial infarction: 1 death
Sederholm et al. [35]	III	170	Lung	Not available	0	Heart failure: 2 grade 3
Cappuzzo et al. [36]	II	117	Lung	6 months	0	Myocardial infarction: 1 death
Sederholm et al. [37]	III	170	Lung	10.5 months	0	Heart failure: 1 grade 3

**Table 3 pharmaceuticals-13-00325-t003:** Selected cardiovascular adverse drug reactions (CV-ADRs, detected as signals) reported for gemcitabine versus the full database from VigiBase, on 1 April 2019. A positive IC_025_ (information component 95% credibility interval lower end) and a reporting odd-ratio (ROR) 95% confidence interval lower-end (95% CI) ≥1 are significant.

Cardiovascular Adverse Events	MedDRA Preferred Term Level	ICSR Reported with Gemcitabine (*n* = 46,898)	ICSR Reported in Full Database (*n* = 18,908,940)	IC (IC_025_)	ROR (95% CI)
Myocardial ischemia	Acute myocardial infarction	69	16,348	0.76 (0.40)	1.71 (1.35–2.17)
*n* = 119	Myocardial ischemia	32	7855	0.70 (0.16)	1.65 (1.17–2.34)
	Acute coronary syndrome	19	4087	0.87 (0.15)	1.88 (1.21–2.95)
Cardiac supraventricular arrhythmias *n* = 308	Atrial flutter	36	4457	1.66 (1.15)	3.28 (2.36–4.55)
	Supraventricular tachycardia	51	7729	1.39 (0.97)	2.67 (2.03–3.52)
	Arrhythmia supraventricular	12	1478	1.59 (0.66)	3.29 (1.86–5.81)
	Atrial tachycardia	9	982	1.69 (0.60)	3.72 (1.93–7.17)
	Atrial fibrillation	199	51,662	0.63 (0.43)	1.56 (1.36–1.79)
	Tachyarrhythmia	12	1771	1.35 (0.42)	2.74 (1.55–4.83)
	Supraventricular extrasystoles	12	2095	1.13 (0.20)	2.32 (1.32–4.09)
Pericardial diseases	Pericardial effusion	151	11,040	2.44 (2.20)	5.59 (4.76–6.57)
*n* = 164	Cardiac tamponade	28	2016	2.37 (1.79)	5.67 (3.90–8.23)
	Pericarditis constrictive	6	171	2.81 (1.44)	14.63 (6.48–33.04)
Heart failure	Cardiac failure	229	40,801	1.17 (0.98)	2.28 (2.01–2.60)
*n* = 484	Cardiac failure acute	15	1914	1.56 (0.74)	3.18 (1.91–5.29)
	Ventricular hypokinesia	10	1131	1.67 (0.64)	3.59 (1.93–6.69)
	Cardiomegaly	42	8196	1.03 (0.56)	2.08 (1.54–2.82)
	Systolic dysfunction	4	226	2.09 (0.35)	7.18 (2.67–19.30)
	Ventricular dysfunction	9	1269	1.38 (0.29)	2.87 (1.49–5.53)
	Cardiac failure congestive	207	63,389	0.40 (0.19)	1.32 (1.15–1.51)
Others	Sinus tachycardia	62	8435	1.54 (1.16)	2.98 (2.32–3.83)
	Atrial thrombosis	7	997	1.34 (0.08)	2.84 (1.35–5.97)

Abbreviations: ICSR: individual case safety report, MedDRA: Medical Dictionary for Regulatory Activities, ROR (95%): reporting odds-ratio and its 95% confidence interval.

**Table 4 pharmaceuticals-13-00325-t004:** Characteristics of patients with gemcitabine-associated cardiovascular adverse drug reactions (CV-ADRs) in VigiBase on 1 April 2019.

Clinical Characteristics		Myocardial Ischaemia *n* = 119	Supraventricular Arrhythmias *n* = 308	Pericardial Diseases *n* = 164	Heart Failure *n* = 484	*p*
Reporting regions	America	67/119 (56%)	195/308 (63%)	**110/164 (67%)**	260/484 (54%)	**0.0003**
	Europe	39/119 (33%)	98/308 (32%)	38/164 (23%)	**198/484 (41%)**	
	Africa	1/119 (1%)	2/308 (1%)	1/164 (1%)	0/484 (0%)	
	Australia	0/119 (0%)	3/308 (1%)	1/164 (1%)	0/484 (0%)	
	Asia	**12/119 (10%)**	10/308 (3%)	14/164 (8%)	26/484 (5%)	
Reporting year	2015–2019	**33/119 (28%)**	65/308 (21%)	42/164 (26%)	121/484 (25%)	**<0.0001**
	2009–2014	**55/119 (46%)**	94/308 (30%)	39/164 (24%)	142/484 (29%)	
	2003–2008	26/119 (22%)	95/308 (31%)	**54/164 (33%)**	121/484 (25%)	
	1997–2002	5/119 (4%)	54/308 (18%)	29/164 (18%)	**100/484 (21%)**	
Reporters	N available	102/119 (86%)	254/308 (82%)	135/164 (82%)	420/484 (87%)	
	Health care professional	99/102 (97%)	236/254 (93%)	122/135 (90%)	394/420 (94%)	0.22
	Other	3/102 (3%)	18/254 (7%)	13/135 (10%)	26/420 (6%)	
Report type	Standard of care	109/119 (91%)	273/308 (89%)	155/164 (94%)	452/484 (93%)	0.06
	Clinical trials	10/119 (9%)	35/308 (11%)	9/164 (6%)	32/484 (7%)	
Sex	N available	105/119 (88%)	295/308 (96%)	152/164 (93%)	451/484 (93%)	
	Male	**65/105 (62%)**	**184/295 (62%)**	57/152 (38%)	**235/451 (52%)**	**<0.0001**
	Female	40/105 (38%)	111/295 (38%)	**95/152 (62%)**	216/451 (48%)	
Age at onset, years	N available	91/119 (76%)	253/308 (82%)	125/164 (76%)	397/484 (82%)	
	Mean (min-max)	65 (23–85)	68 (32–85)	**55 (16–81)**	64 (20-89)	**<0.0001**
	Standard deviation	11.5	9.4	15.6	11.7	
Suspected drugs	Gemcitabine alone	33/119 (28%)	**178/308 (55%)**	**124/164 (76%)**	**292/484 (60%)**	**<0.0001**
	Gemcitabine and ≥1 other	**86/119 (72%)**	130/308 (45%)	40/164 (24%)	192/484 (40%)	
Other concomitant or suspected drugs	Taxanes	29/119 (24%)	**122/308 (40%)**	34/164 (21%)	99/484 (21%)	**<0.0001**
	Vinca alkaloids	7/119 (6%)	18/308 (6%)	9/164 (5%)	36/484 (7%)	0.74
	Anthracyclines	8/119 (7%)	8/308 (3%)	5/164 (3%)	20/484 (4%)	0.22
	Topoisomerase I inhibitors	2/119 (2%)	7/308 (2%)	4/164 (2%)	5/484 (1%)	0.22
	Platins	**55/119 (46%)**	80/308 (26%)	25/164 (15%)	104/484 (21%)	**<0.0001**
	Antimetabolites	10/119 (8%)	32/308 (10%)	13/164 (8%)	36/484 (7%)	0.94
	Mustard gas derivative	2/119 (2%)	3/308 (1%)	3/164 (2%)	11/484 (2%)	0.63
	Angiogenesis inhibitors	16/119 (14%)	28/308 (9%)	10/164 (6%)	39/484 (8%)	0.16
	Human epidermal growth factor receptor 2 blockers	4/119 (3%)	4/308 (1%)	20/164 (1%)	**29/484 (6%)**	**0.02**
	Epidermal growth factor receptor blockers	**19/119 (16%)**	22/308 (7%)	11/164 (7%)	26/484 (5%)	**0.001**
	Immune checkpoint inhibitors	0/119 (0%)	3/308 (1%)	**5/164 (3%)**	0/484 (0%)	**0.001**
Duration of administration, days	N available	47/119 (40%)	135/308 (44%)	47/164 (29%)	213/484 (44%)	**<0.0001**
	Median	28	16	60	91	
	Interquartile range	7-97	1-57	7-146	20–176	
Time to onset, days	N available	46/119 (39%)	142/308 (46%)	62/164 (38%)	202/484 (42%)	**<0.001**
	Median	29	14	55	75	
	Interquartile range	6-80	4-52	13–144	22–166	
Severe adverse events *		**113/119 (94%)**	240/308 (78%)	129/164 (79%)	369/484 (76%)	**<0.0001**
Recovery	N available	51/119 (43%)	132/308 (43%)	55/164 (34%)	163/484 (34%)	**0.02**
	Recovered	35/51 (69%)	**107/132 (81%)**	**48/55 (87%)**	115/163 (70%)	
	Not recovered or sequelae	**16/51 (31%)**	25/132 (19%)	7/55 (13%)	**48/163 (30%)**	
Indications	N available	93/119 (78%)	228/308 (74%)	97/164 (59%)	293/484 (61%)	**<0.0001**
	Pancreatic cancer	**31/93 (33%)**	**64/228 (28%)**	**24/97 (25%)**	**115/293 (39%)**	
	Lymphoma	4/93 (4%)	17/228 (7%)	**19/97 (20%)**	16/293 (6%)	
	Lung cancer	**24/93 (26%)**	**83/228 (36%)**	**23/97 (24%)**	**70/293 (24%)**	
	Urothelial cancer	**13/93 (14%)**	14/228 (8%)	6/97 (6%)	24/293 (8%)	
	Breast cancer	5/93 (5%)	19/228 (8%)	8/97 (8%)	**36/293 (12%)**	
	Ovarian cancer	6/93 (6%)	8/228 (3%)	8/97 (8%)	12/293 (4%)	
	Bile duct cancer	8/93 (9%)	**21/228 (9%)**	2/97 (2%)	13/293 (5%)	
	Sarcoma	2/93 (2%)	2/228 (1%)	7/97 (7%)	7/293 (2%)	

* A severe ADR was defined as causing death; being life-threatening; requiring hospital stay (initial or prolonged); or leading to persistent or clinically significant disability, congenital anomaly, birth defect, or any other medically important conditions. Bold is defined as statistically significant (*p* < 0.05).

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
