# Peer review of "Cardiotoxicity Associated with Gemcitabine: Literature Review and a Pharmacovigilance Study"

_pharmaceuticals, 2020, doi:10.3390/ph13100325_

Round 1

Reviewer 1 Report

In this paper the authors aim to better define the cardiotoxicity induced by gemcitabine through three steps namely,

  1. description of three clinical cases, retrived from UNICO-GRECO Cardio-oncology program related to patients who developed a cardiovascular ADR likely related to gemcitabine;
  2. literature review through Medline
  3. Pharmacovigilance study using VigiBase as reporting database. The study is well written, easy to read and with very interesting results. With regard to pharmacovigilance study, I really appreciated the Information Component as method for disproportionality evaluation in Vigibase. However, in order to reletivize your results, is it possible to complete your analysis with a comparison between gemcitabine and, for example, doxorubicine? Of course, for this purpose the reporting odds ratio (ROR) could be the most suitable disproportionality measure.

Author Response

Paris, 4th October 2020

Dear Reviewer,

We would like to thank the reviewer for its helpful comments and suggestions.

You will find attached a revised version of our manuscript with tracked-in-blue changes in the body of the text and you will find the response to your comments.

In this paper, the authors aim to better define the cardiotoxicity induced by gemcitabine through three steps namely,

  1. description of three clinical cases, retrieved from UNICO-GRECO Cardio-oncology program related to patients who developed a cardiovascular ADR likely related to gemcitabine;
  2. literature review through Medline
  3. Pharmacovigilance study using VigiBase as reporting database. The study is well written, easy to read and with very interesting results. With regard to pharmacovigilance study, I really appreciated the Information Component as method for disproportionality evaluation in Vigibase. However, in order to relativize your results, is it possible to complete your analysis with a comparison between gemcitabine and, for example, doxorubicin? Of course, for this purpose the reporting odds ratio (ROR) could be the most suitable disproportionality measure.

We agree that disproportionality can be calculated by either the information component (IC) or the reporting odds ratio (ROR) when using the full database as the comparator. In order to moderate our results, as suggested, we completed our analysis by adding the ROR with 95%CI in Table 3.

We also made a detailed check for language errors throughout the manuscript.

We hope that these revisions improve the manuscript such that you deem it suitable for publication in Pharmaceuticals.

Once again, we thank the reviewer for its interest in our work.

Best regards,

Joe-Elie Salem

Reviewer 2 Report

Dear Editor,

the manuscript entitled with ‘Cardiotoxicity associated with gemcitabine: case series, literature review and a pharmacovigilance study’ aims to investigate for cardiovascular events in cancer patients, treated with gemcitabine. The authors have also included a case series of three representative cases and their management. The study has many obvious limitations, such as the retrospective data analysis, risk of under-reporting or lack of mechanistic insides. However, gemcitabine associated toxicity is an important question in daily clinical practice and the authors did an excellent job to deliver a comprehensive analysis of available data on potential harmful side effects of gemcitabine. These data are important to be published and they are interesting for a broad readership. This analysis could further be the basis for a prospective evaluation of gemcitabine associated cardiotoxicity.

There are some minor issues that need to be addressed:

  • Patient 2: did they perform a cardiac catheter or pericardiocentesis?
  • Figure 1:
    • CT-scans: The authors should comment if it is the arterial phase (patient 1?) and if it’s a native CT-scan in p2 and p3. Why?
    • Figure legend should provide major findings of CT-scans.
    • Are there movies from echocardiography available?
    • Diagnosis of underlying cancer should be included (type, timepoint).
    • Labeling should be improved (TNT or TNI?)
    • ECG should be included (at least the description)
    • Are there strain data available?
    • Duration and dose of gemcitabine should be provided in the table.
  • Figure 2:
    • Total number of all adverse events should be shown
    • The authors should consider including all sub-classifications (e.g. AF, VF) from table 2 into the figure
  • To improve readability, the authors should consider providing the essential findings in a smaller number of figures/tables and moving some of the data into supplemental figure/tables (e.g. table 2, table 4)
  • In table 4, SVA needs to be explained in the legend.
  • The authors should speculate on surveillance strategies. Who needs to be seen by a cardiologist and when do I only need to do an ECG?

Author Response

Paris, 4th October 2020

Dear Reviewer,

We would like to thank the reviewer for its very helpful comments and suggestions.

You will find attached a revised version of our manuscript with tracked-in-blue changes in the body of the text and you will find the point-by-point responses to your comments.

The manuscript entitled with ‘Cardiotoxicity associated with gemcitabine: case series, literature review and a pharmacovigilance study’ aims to investigate for cardiovascular events in cancer patients, treated with gemcitabine. The authors have also included a case series of three representative cases and their management. The study has many obvious limitations, such as the retrospective data analysis, risk of under-reporting or lack of mechanistic insides. However, gemcitabine associated toxicity is an important question in daily clinical practice and the authors did an excellent job to deliver a comprehensive analysis of available data on potential harmful side effects of gemcitabine. These data are important to be published and they are interesting for a broad readership. This analysis could further be the basis for a prospective evaluation of gemcitabine associated cardiotoxicity.

There are some minor issues that need to be addressed:

  • Patient 2: did they perform a cardiac catheter or pericardiocentesis?

Patient 2 did not benefit from pericardiocentesis. Cardiac catheter was performed few weeks after cardiac decompensation and did not show any significant lesion. As suggested by Reviewer 3, we shortened part relating to the case-seriest.

We also followed the Reviewer’s recommendation by removing this part from the Results and Methods sections. We added the details of these case-reports in Table 1 and use them as initiators for our work by summarizing in the text:

Line 57: “We identified 3 patients in Saint-Antoine and Pitié-Salpêtrière hospitals (Paris, France) who developed a CV-ADR suspected to be related to gemcitabine. They presented with pancreatic cancer and CV-ADR were pericardial effusion associated with heart failure in two of them, which occurred 6 to 8 months after gemcitabine initiation. Partial or complete recovery was observed after instauration of heart failure therapeutics (see details of in Table 1). One patient presented its pericardial effusion as part of a CLS recovered after glucocorticosteroids administration (Supplementary Figure 1 presents dynamics of his recovery).”

  • Figure 1 :

CT-scans: The authors should comment if it is the arterial phase (patient 1?) and if it’s a native CT-scan in p2 and p3. Why? Figure legend should provide major findings of CT-scans.

Indeed, we clarified the fact that Figure 1 summarizes our third case report. We are sorry for this misunderstanding and changed the Figure Title: “Summary of one of our case-report presenting a capillary leak syndrome associated with gemcitabine reversible after glucocorticoids.”

We also added the phase of the CT-scan and the major findings in the Figure Legend  :” As illustrated on the CT-scan images (arterial phase), pleural and pericardial effusion progressively disappeared in less than 10 days after onset.

Are there movies from echocardiography available?

Indeed, movies from echocardiography are available (admission, day 5, day 10) and have been uploaded as Supplementary Data.

Diagnosis of underlying cancer should be included (type, timepoint). Duration and dose of gemcitabine should be provided in the table.

As suggested, we added these points into the Figure Legend: “The patient had a metastatic pancreatic carcinoma and the adverse event started 8 months after diagnosis (corresponding to 8 cycles of gemcitabine with a total dose of 24 000mg/m²).”

Labeling should be improved (TNT or TNI?)

ECG should be included (at least the description)

Are there strain data available?

Following your suggestions, we added these elements (TNI, ECG description and strain data) in the Figure (now supplementary Figure 1).

  • Figure 2:

Total number of all adverse events should be shown

We modified this Figure in order to show the total number of all adverse events.

The authors should consider including all sub-classifications (e.g. AF, VF) from table 2 into the figure

Figure 2 is a Venn diagram which is the most visual way to show our results, albeit non-exhaustive Indeed, because of the high number of sub-classifications (n=22), including them into the Venn diagram was not feasible.

To remedy this, we made a more exhaustive graphical representation, an UpsetR graph (Supplementary Figure 2) which includes all sub-classifications from Table 2.

  • To improve readability, the authors should consider providing the essential findings in a smaller number of figures/tables and moving some of the data into supplemental figure/tables (e.g. table 2, table 4)

We believe that Table 2 is the essential table of our work by showing the results of the disproportionality analysis. However, we agree that Table 4 can be transferred into Supplementary Data. We also transferred the Figure 1 into Supplementary Data because Reviewer 3 suggested to shorten the case-series part.

  • In table 4, SVA needs to be explained in the legend.

We changed SVA for “Supraventricular arrhythmias” in Table 4.

  • The authors should speculate on surveillance strategies. Who needs to be seen by a cardiologist and when do I only need to do an ECG?

Our work allowed to better characterize the cardiovascular events associated with gemcitabine, notably the clinical characteristics including time to onset and severity. However, the surveillance strategies including the cardiovascular evaluation (ECG, echocardiography, troponin) should be evaluated prospectively in a separate study.

We added the following part on patient management in the Discussion:

Line 189: “Although CV-ADRs were severe, required hospitalization and often were not fully reversible, cardiac toxicity of gemcitabine is not well known to clinicians. Even in the absence of cardiovascular risk factors, cardiotoxicity related to gemcitabine should be immediately suspected in the case of breathlessness, palpitations, or chest discomfort. Moreover, cardiovascular screening (e.g. echocardiography assessing pericardium and left ventricular ejection fraction, past cardiac history and prior radiation therapy) may identify patients at higher risks of gemcitabine-associated cardiotoxicity but this strategy needs to be evaluated in a dedicated study. Furthermore, in some case-reports (Table-1), gemcitabine re-challenge led to recurrence of the CV-ADR suggesting that decision to restart gemcitabine needs to be weighted between cardiotoxicity and anti-tumor efficacy.[42]

We also made a detailed check for language errors throughout the manuscript.

We hope that these revisions improve the manuscript such that you deem it suitable for publication in Pharmaceuticals.

Once again, we thank the reviewer for his interest in our work.

Best regards,

Joe-Elie Salem

Reviewer 3 Report

MDPI: Pharmaceuticals

Cardiotoxicity associated with gemcitabine: case series, literature review and a pharmacovigilance study

This study outlines the considerable risk of severe cardiotoxicity by treatment of patients with various cancers with gemcitabine (GCB). The results of this review potentially are of high clinical relevance not only for the pharmacologist but even more for the clinician.

Unfortunately, the applied methodology may not be sufficient to adequately evaluate the cardiovascular risk associated with GCB-application. Moreover, in the present form the manuscript is difficult to read, and the methods applied have not clearly been presented. This will be outlined in the following:

  • FDA-label: the FDA judgement of a low cardiovascular risk treatment with GCP of is a major background information. The assumption of an increased cardiovascular risk not mirrored by the FDA report principally would arise the need of official re-evaluation of this substance. In addition, the judgment of the European authorities (EMA) should be outlined and discussed in addition.
  • Case reports: the case reports are interesting to read but remain anecdotal. They do not really increase the scientific evidence level, and therefore their text should markedly be shortened and only serve as initiator for the systematic review within the introduction.
  • Literature review: the literature review only includes case reports but no clinical trial. This potentially could be a serious draw-back, as case reports may be confounded by a variety of factors including selective reporting. Therefore a structured review of clinical studies (RCT, controlled cohort studies, registries) should be the basis of this evaluation. This basis may be supplemented by case reports.
  • Pharmacovigilance study: a disproportionality analysis has been performed. By this way the incidence of a specific ADR occurring during gemcitabine therapy could be compared with populations not treated with this drug. This appears to be the only valid approach presented in this paper for estimating gemcitabine ADRs, and it would be more informative to concentrate on this data base.
  • Still there remain further questions:
    • How do these data compare with the controlled clinical studies on gemcitabine being published so far? Please consider all controlled trials being published and critically compare these results with the data presented here in this manuscript. Potential differences then need to be discussed.
    • In case that there is a suspected underreporting of GCB side effects in the past, the authors need to clearly show this by data and give a clear statement.
    • Are there any data suggesting adverse interactions between gemcitabine and other concomitant drugs? This is important as some drugs may aggravate GCB side effects
    • Which are the potential alternatives to gemcitabine, and how do they compare with respect to effects and adverse side effects?
    • Side effects of GCB may depend on pre-existing cardiovascular diseases. Are there any data supporting this hypothesis?

Taking together, reporting of severe cardiovascular side effects of GCB treatment is important. However, the actual presentation should be strengthened and concentrate on data that allow a realistic estimation of case incidence, potential effectors and confounders as well as special risk factors and pre-existing risk diseases facilitating GCB cardiovascular side effects. If possible data also may be compared with other drugs being applied within the clinical indications of interest.

Author Response

Paris, 4th October 2020

Dear Reviewer,

We would like to thank you for your very helpful comments and suggestions.

We attach a revised version of our manuscript with tracked-in-blue changes in the body of the text and you will find the point-by-point responses to your comments. We feel the revised manuscript clarifies a number of issues and hope it will be suitable for publication in your esteemed journal.

Once again, thank you for your time.

Best regards,

Joe-Elie Salem, MD, PhD

This study outlines the considerable risk of severe cardiotoxicity by treatment of patients with various cancers with gemcitabine (GCB). The results of this review potentially are of high clinical relevance not only for the pharmacologist but even more for the clinician.

Unfortunately, the applied methodology may not be sufficient to adequately evaluate the cardiovascular risk associated with GCB-application. Moreover, in the present form the manuscript is difficult to read, and the methods applied have not clearly been presented. This will be outlined in the following:

  • FDA-label: the FDA judgement of a low cardiovascular risk treatment with GCP of is a major background information. The assumption of an increased cardiovascular risk not mirrored by the FDA report principally would arise the need of official re-evaluation of this substance. In addition, the judgment of the European authorities (EMA) should be outlined and discussed in addition.

As suggested, we added the EMA judgement in addition to the FDA judgement to reinforce the background information:

Line 54: “The European Medicines Agency also notifies the particular caution with patients presenting a history of cardiovascular events due to the risk of CV-ADR with gemcitabine.[6]”

  • Case reports: the case reports are interesting to read but remain anecdotal. They do not really increase the scientific evidence level, and therefore their text should markedly be shortened and only serve as initiator for the systematic review within the introduction.

We followed the recommendation of the reviewer by removing this part from the Results and Methods sections. We added the details of these case-reports in Table 1 and used them as premises for our work by the following paragraph:

Line 57: “We identified 3 patients in Saint-Antoine and Pitié-Salpêtrière hospitals (Paris, France) who developed a CV-ADR suspected to be related to gemcitabine. They presented with pancreatic cancer and CV-ADR were pericardial effusion associated with heart failure in two of them, which occurred 6 to 8 months after gemcitabine initiation. Partial or complete recovery was observed after instauration of heart failure therapeutics (see details of in Table 1). One patient presented its pericardial effusion as part of a CLS recovered after glucocorticosteroids administration (Supplementary Figure 1 presents dynamics of his recovery).

  • Literature review: the literature review only includes case reports but no clinical trial. This potentially could be a serious draw-back, as case reports may be confounded by a variety of factors including selective reporting. Therefore a structured review of clinical studies (RCT, controlled cohort studies, registries) should be the basis of this evaluation. This basis may be supplemented by case reports.

In order to strengthen our literature review, we performed a systematic search on Pubmed regarding clinical studies evaluating gemcitabine as monotherapy, for which we added inclusion criteria of the retained clinical trials in the Methods section, in the Supplementary Figure 3 and in line 246:

We also performed a systematic search on Pubmed (MEDLINE) for randomized clinical trials using the term “gemcitabine” up to September 30, 2020 and found 958 manuscripts. We then selected articles evaluating gemcitabine monotherapy in at least one of the treatment arms and found 106 studies (Supplementary Figure 3).”

We now showed these results in a new Table (Table 2) and in line 88:

Moreover, among the 106 randomized clinical trials evaluating gemcitabine as monotherapy (overall number of patients=14 015), 17 trials (n=2 386) had reports for at least one CV-ADR in the published work (Supplementary Figure 2 for the flow chart of selected trials). In these latter 17 trials, 33 CV-ADR were reported, leading to an estimate of CV-ADR incidence rate on gemcitabine monotherapy ranging from 0.24% (33/14015) to 1.38% (33/2386), varying with the denominator considered. Quality of CV-ADR reporting in these trials, mostly from decades ago was too low to make sure that all CV-ADR events were effectively captured and reported in the publication. These events were severe in n:27/33 (82%) (n:14/33 grade 3/4 and n:13/33 death), and included 8 MI, 2 pericardial effusions, 7 HF and 1 arrhythmia (Table-2).”

Pharmacovigilance study: a disproportionality analysis has been performed. By this way the incidence of a specific ADR occurring during gemcitabine therapy could be compared with populations not treated with this drug. This appears to be the only valid approach presented in this paper for estimating gemcitabine ADRs, and it would be more informative to concentrate on this data base.

Still there remain further questions:

  • How do these data compare with the controlled clinical studies on gemcitabine being published so far? Please consider all controlled trials being published and critically compare these results with the data presented here in this manuscript. Potential differences then need to be discussed.

Following your previous query, we reported clinical studies on gemcitabine monotherapy following a systematic search in the new Table 2 and in the Results section.

We also discussed these new findings, line 176:

Gemcitabine was associated with MI, SVA, HF and pericardial diseases. The same cardiotoxicity signals from gemcitabine have also been reported in clinical trials. Aapro et al. pooled 979 patients treated by gemcitabine in 22 phase-2 trials and showed that incidence of MI, HF, arrhythmias and pericarditis were 0.5%, 0.4%, 0.2% and 0.1%, respectively.[38] Our meta-proportion analysis of randomized clinical trials evaluating gemcitabine as monotherapy showed similar incidence estimates of ~1% for overall CV-ADR including MI, SVA, HF and pericardial diseases. We showed that mortality associated with these CV-ADRs ranged from 3% for after pericardial diseases versus 17% for after MI and HF. Severity including death and grade 3/4 events reported in randomized clinical studies (82%) was similar to that of the pharmacovigilance reports (87%). Though, it has to be noted that the quality of CV-ADR adjudication in the trials considered in our analysis was low precluding the possibility of precisely characterizing these CV-ADR. Pericardial diseases had the strongest association with gemcitabine administration (highest IC025, Table-3), and were presented as cardiac tamponade (28/164, 17%) or constrictive pericarditis (6/164, 4%). Noteworthy, pericardial disorders could be part of CLS, a systemic disease determined by vascular protein leakage and diffuse serosa effusions requiring specific management, including glucocorticoids.[39-41] Association between gemcitabine and CLS using VigiBase have been described previously.[5] »

  • In case that there is a suspected underreporting of GCB side effects in the past, the authors need to clearly show this by data and give a clear statement.

Indeed, this a known limitation for which we added a statement supported by references:

Line 225: “While the accurate magnitude of underreporting cannot be computed, estimates vary up to 90% of the actual adverse events not being reported.[47,48]

  • Are there any data suggesting adverse interactions between gemcitabine and other concomitant drugs? This is important as some drugs may aggravate GCB side effects

It has to be noted that for a majority of patients (64%), gemcitabine was the only suspected drug for the reported cardiotoxicity. However, we agree that concomitant drugs may aggravate gemcitabine CV-ADR, which is why we presented these data in Table 3 and added a paragraph in the Results section:

Line 140: “Prevalence of other concomitant anticancer drugs intake as a function of type of gemcitabine associated CV-ADRs are presented in Table-4. Combination therapy (gemcitabine and at least one other anticancer drug suspected) was more incriminated than gemcitabine alone in myocardial infarction reports (86/119, 72% vs. 33/119, 28%). A significant association was detected for HER-2 blockers and HF (p=0.02), taxanes and SVA (p<0.0001), immune-checkpoint-inhibitors and pericardial disorders (p=0.001), epidermal growth factor receptor blockers and/or platins and MI (p=0.001) – see Table-4 for more details.”

Taking together, reporting of severe cardiovascular side effects of GCB treatment is important. However, the actual presentation should be strengthened and concentrate on data that allow a realistic estimation of case incidence, potential effectors and confounders as well as special risk factors and pre-existing risk diseases facilitating GCB cardiovascular side effects.

We thank the reviewer for this insightful comment. Unfortunately, incidence cannot be reported through a disproportionality analysis unless having all post-marketing gemcitabine data. This limit is specified in the Discussion line 232: “The exact denominator of patients exposed to gemcitabine cannot be evaluated precluding estimation of incidence of these CV-ADRs using VigiBase.”

We also provided data from literature concerning if pre-existing risk diseases were associated with gemcitabine CV-ADR (see below).

  • If possible data also may be compared with other drugs being applied within the clinical indications of interest. Which are the potential alternatives to gemcitabine, and how do they compare with respect to effects and adverse side effects?

Gemcitabine is widely used, either alone or in combination for the treatment of multiple cancers including lung, pancreas, bladder, breast, ovary and bile duct carcinomas, lymphomas and uterine sarcomas. Many alternatives such as taxanes, vinca-alkaloids or antimetabolites can be proposed but gemcitabine is generally preferred in elderly or fragile patients due to its lower toxicity profile versus the other possible alternative anticancer drugs. Thus, gemcitabine re-challenge after a suspected induced CV-ADR led to recurrence of the CV-ADR in most case-reports suggesting that non-cardiotoxic alternative chemotherapy should probably be discussed in these latter cases.

We added the following part on this point in the Discussion:

Line 195: “Furthermore, in some case-reports (Table-1), gemcitabine re-challenge led to recurrence of the CV-ADR suggesting that decision to restart gemcitabine needs to be weighted between cardiotoxicity and anti-tumor efficacy.[42]”

  • Side effects of GCB may depend on pre-existing cardiovascular diseases. Are there any data supporting this hypothesis?

Vigibase is not appropriate to determine if CV-ADRs of gemcitabine may depend on pre-existing cardiovascular diseases because these data regarding co-morbidities are often missing. Thus, we can provide the following data from literature:

- For heart failure: A retrospective study suggested diabetes, coronary artery disease and a total gemcitabine dose>17.000mg/m2 as risk factors for developing gemcitabine-induced heart failure but validity of these results are limited by a small sample size (7 heart failure on a total of 156 gemcitabine treated patients). (doi:10.1200/jco.2004.10.112.)

- For myocardial infarction: Similar to fluoropyrimidines (other antimetabolites), vasospasm has been proposed to be responsible for gemcitabine-associated myocardial infarction. Interestingly, myocardial infarction associated with gemcitabine were often co-reported with strokes in our study, suggesting that shared cardiovascular risk factors may play a role.

- For pericarditis: Vogl et al. highlighted an association between gemcitabine and pericarditis via a radiation recall reaction suggesting that a previous cardiac irradiation is a risk factor for developing a pericarditis on gemcitabine. ( doi:10.1080/10428190500158649.)

- Overall, patients can be predisposed to gemcitabine cardiotoxicity (pre-existing cardiovascular diseases, previous cardiac irradiation, combination with other cardiotoxic anti-cancer drugs) but these CV-events on gemcitabine can also occur in the absence of any predisposing factors.

We added the following to the discussion to assess this point raised by the reviewer:

Line 189: “Although CV-ADRs were severe, required hospitalization and often were not fully reversible, cardiac toxicity of gemcitabine is not well known to clinicians. Even in the absence of cardiovascular risk factors, cardiotoxicity related to gemcitabine should be immediately suspected in the case of breathlessness, palpitations, or chest discomfort. Moreover, cardiovascular screening (e.g. echocardiography assessing pericardium and left ventricular ejection fraction, past cardiac history and prior radiation therapy) may identify patients at higher risks of gemcitabine-associated cardiotoxicity but this strategy needs to be evaluated in a dedicated study.”

We also made a detailed check for language errors throughout the manuscript.

Round 2

Reviewer 3 Report

Dear Professor Salem, dear authors,

the manuscript has been significantly improved. Regarding the clinical importance of the presented observations, I think the manuscript is ready for publication.

with kindest regards

Bernhard Rauch